# Toward Blockchain Realization

**Chih-Wen Hsueh** [1,*] 🔘 **and Chi-Ting Chin** [2]

1   Department of Computer Science and Information Engineering, National Taiwan University, Taipei 10617, Taiwan
2   Department of Risk Management and Insurance, Ming Chuan University, Taipei 11103, Taiwan; debbyjin@zeta.mcu.edu.tw
*   Correspondence: cwhsueh@csie.ntu.edu.tw; Tel.: +886-928-777-371

**Abstract:** Since FinTech was stimulated by the invention of blockchain, without the full realization of blockchain technologies in the following years, FinTech has not been fully realized. We discuss some myths and reasons for why blockchain technologies were not fully realized. The lack of distributed synchronization might be the most difficult challenge such that the trust provided by blockchain is not good enough for public use. We propose a mathematical solution with a new consensus mechanism based on general Proof-of-Work mining, called Proof-of-PowerTimestamp, to reach distributed synchronization and reduce power consumption to less than one billionth of Bitcoin. We also discuss related issues toward blockchain realization once the distributed synchronization and energy consumption problems are solved. Since the issues are mostly interdisciplinary or multidisciplinary, researchers are invited to cooperate to help blockchain realization as soon as possible.

**Keywords:** FinTech; blockchain; finality; consensus; Proof-of-Work; energy consumption; realization

## 1. Introduction

Financial technology (abbreviated FinTech ) is the technology and innovation that aims to compete with traditional financial methods in the delivery of financial services. It is an emerging industry that uses technology to improve activities on automating investments, insurance, trading, banking services and risk management. Most technologies involve artificial intelligence (AI), Big Data, robotic process automation (RPA) and blockchain [1]. FinTech history dates back to the 19th century and even before that [2], but it was stimulated the most in the past two decades by blockchain because of the fundamental change of currencies and payment methods.

The words block and chain were used separately in Bitcoin white paper [3], in 2008, to devise electronic cash on the Internet. Before it was introduced in *The Economist* edition on 31 October 2015 as a trust machine, blockchain became the most significant innovation since the Internet [4]. Bitcoin was the first blockchain application before the term blockchain was coined. In addition to support Bitcoin-like cryptocurrencies, blockchain has supported more than 15,000 cryptocurrencies by December 2021 [5]. People has also developed many new features for blockchain where trust needs to be enforced in various applications, such as smart contract, which is a trustworthy program verified by peer nodes. Although there are still problems such as energy consumption, performance, security, scalability, privacy, storage and so on that still need to be solved, the total market value of cryptocurrencies has reached 2.235T USD on 31 December 2021 [6], about 2.35% of the 2021 global GDP (Gross Domestic Product) 94.935T USD [7]. The percentage is not high because, in order for value to be transferred on Internet, the trust and finality supported by current blockchains are still only stochastic. Stochastic finality might be good enough for speculators but not for prevalent applications for the public, such as Central Bank Digital Currency (CBDC). On the other hand, the percentage is very high because it ranks 10th among the GDP of all countries in 2021. In addition, the other problems mentioned above mainly come from the

fact that there is no global event ordering [8] for synchronization in distributed systems. Since trust can reach a consensus in a timely manner, with distributed synchronization, blockchain can support real trust for most Internet applications, let alone the metaverse. Without trust supported by blockchain, the metaverse is only an augmented online game.

To reach consensus in computer science is to make participating processes agree on some data value or situation. Consensus plays the most important role for the behavior or features supported in blockchain. There is a wide variety of research and applications that attempt to solve different problems by using different methods to reach a consensus. More definitions of basic terms used in blockchain can be found in an analysis paper [9]. As shown in Table 1 of the analysis paper [9], efforts on public blockchain can only achieve probabilistic (or stochastic) finality with huge energy consumption or unfair capitalism, and efforts on permitted applications can achieve absolute finality, but they are private blockchains and are not trustworthy enough. The situation looks quite different from the figures describing how different consensuses work. Basically, they all first collect data from peer nodes and then decide which data to keep, or they form a situation with data to reach consensus and grow their blockchains. However, a consensus cannot be reached at once in a distributed system, resulting partitions [10]. Although none of the instances solve all problems, they own enough users or markets to survive with different advantages, regardless of the notorious energy consumption in public blockchains or possible collusions in private blockchains. Actually, most of the problems come from the fact that stochastic finality cannot provide distributed synchronization efficiently and effectively. Common solutions are longer wait times or using multiple resources that ignore possible attacks or problems in real use, usually with various exceptions and a large number of transactions. Using epoch and several synchronized block intervals, deterministic finality can be achieved [11] with more flexibility than absolute finality. However, it requires long (100-block) epoch times and utilizes a distributed hash table to assume perfect (under some bound) network connection [12]. In reality, O(1) complexity is difficult to achieve in a jittering network environment, and it might not be flexible and trustworthy enough to impose business logic or reflect user investments.

As long as there are some stochastic and speculative benefits, applications do not need to wait for 100% mature technologies to develop and users would like to take some risks or even ignore risks. Social research studies even design laws and policies for governments to help reduce energy consumption [13], which might bring inconvenience in daily lives. Sedlmeir et al. [14] even found reasons that the energy consumption of blockchain does not pose a large threat to climate from a holistic point of view, while keeping the advantages of blockchain. Innovation occurs in this interaction of research and application but myths and abuses are incurred as well. The myth might be another reason for delaying realization. Moreover, although realized blockchain might bring a lot of benefits, vested interests or malicious authorities might go against it. Few research has addressed the problem of how to solve complex, technical, social, mental and business logical problems above. If we can find a consensus mechanism with flexible deterministic finality so that distributed synchronization is possible for blockchain transactions with various business logic, the problems above might be solvable. We can also simplify the complexity of understanding blockchains in this perspective instead of following various technologies with temporary solutions.

OurChain [15] based on Bitcoin PoW (Proof-of-Work) was prototyped on July 2019 as a public blockchain with consensus on EPoW (Estimable PoW) [16], reaching a deterministic finality with a two-block epoch and global event ordering, called Proof-of-PowerTimestamp (PoPT), by comparing the timestamp first and then estimating computing power. Each block interval is 2 s. However, OurChain was not published because EPoW is not generalized pr simple enough, resulting in a situation where the distributed synchronization problem was not solved. Extending from EPoW, GPoW (General PoW) is a general PoW supported by the order statistic [17,18] model. GPoW constructs a trust model of blockchain with closed-form formula to any given level of coefficient variance, develops the distributed syn-

chronization solution and solves the energy consumption problem of PoW. The trilemma, consistency, availability and partition tolerance in a distributed system [10] becomes feasible by synchronization in a real-time manner. The closed-form formula can help to adjust system behavior dynamically to adapt to network jitter in real-time for reaching distributed synchronization. By GPoW, a flexible deterministic finality can be achieved by dynamically adjusting the target value of mining difficulty for different user behavior. The user behavior can be also modeled by functions of estimated computing power economically and flexibly by using EPoW. Our contribution in this paper can be summarized as follows:

1. We first present PowerTimestamp and the Proof-of-PowerTimestamp consensus mechanism supporting deterministic finality of a two-block epoch and decentralized global event ordering so that distributed synchronization is feasible.
2. We present some myths and corrections on blockchain. In particular, the most important reason why blockchain is trustworthy is the decentralization supported by the randomness from hash-based PoW consensus mechanism.
3. We first present OurChain, a public blockchain with deterministic finality based on Bitcoin with smart contract and PoPT consensus of mining on EPoW.
4. We first propose that the GPoW mining can be modeled by order statistic so that a general PoW mining can be constructed with the deterministic behavior of coefficient variance forming a closed-form formula such as a trust indicator for blockchain for any given level of precision.
5. With GPoW, we mathematically show that energy consumption from mining can be less than one billionth of Bitcoin.
6. To quickly figure out the closed-form formula from a series sum of products with binomial coefficients, we invent a new technique of simplification from differences, which is called accumulation.
7. For promoting open source with more fairness and justice, we propose the Benevolence License [19] and apply it on all technologies.

Section 2 introduces EPoW and discusses some traditional concepts and myths. GPoW is introduced in Section 3. Other realization issues are discussed in Section 4. The paper is concluded in Section 5. Appendix A introduces accumulation.

## 2. Background

Since blockchain is an emerging technology, there are still different descriptions and myths on the features of blockchain. Even though there is no definite answer for questions such as why blockchain is trustfworthy or how much it can be trusted, we attempt to answer this in this section briefly. The following descriptions are not really a background, because they have been used in lectures or talks for some years but are not yet published. We did not publish it because we did not find the mathematical model of GPoW to really know what consensus and trust are in hash-based PoW blockchain. In addition, only a prototype of OurChain is implemented. We will publish them in detail as soon as possible and implement OurChain with full features for proof as well.

### 2.1. Common Features of Blockchain

Decentralized, immutable, transparent and anonymous characteristics might be the most common features mentioned with blockchain. However, there are still related myths to be corrected.

- **Decentralized**: It is the most important source of trust in blockchain, especially with the permissionless control of mining nodes. With the hash-based PoW, a block from any node can be verified by all nodes and be randomly chosen to be appended as the latest block. A trustless trust is the most trustworthy. However, this does not mean it removes centralization at all. Some authority as a center might still be necessary, such as the government or a central bank, to control the interest or tax rates;
- **Immutable**: The data are reliable because they are hash-chained and are very difficult to change once written. However, it is not trustworthy since wrong inputs might

produce wrong outputs and it might be replaced for the consensus mechanism, e.g., the longest chain policy in PoW. Therefore, appending transactions to offset previous mistakes is inevitable and acceptable;

- **Transparent**: All data in most blockchains are not encrypted and transparent to anyone. It might be misunderstood because the translation of cryptocurrency is wrong and is encrypted, such as 加密貨幣 in Chinese. Actually, 密碼貨幣 means that the term "cryptographic" is more appropriate and also the example adopted in Wikipedia;
- **Anonymous**: The block ID or transaction ID is only a hash of itself. It is anonymous but is still traceable by the IP address of network packet or other detective methods. Actually, anything used should leave some clues for being traced unless it is not used anymore.

More features and myths are described in the following subsections.

### 2.2. Consensus

To reach a consensus in distributed computing is a result of coordinating processes that agree on some data value needed during computation to achieve overall system reliability in the presence of a number of faulty processes. It is a fundamental problem in distributed systems with many solutions for different scenarios. In other words, there might be no general solution. In a fully asynchronous message-passing distributed system, in which at least one process may have a crash failure, it has been proven in FLP impossibility [20], resulting the fact that a deterministic algorithm for achieving consensus is impossible. While real world communications in systems are often inherently asynchronous, it is more practical and often easier to model them as synchronous systems, given that an asynchronous system naturally involves more issues to solve than a synchronous one [21]. The consensus mechanism plays the most important role in blockchain to support trust. However, with the requirements of decentralization, it is not a trivial problem in public blockchain, where permissionless and asynchronous blockchains are natural because anyone can participate at any time without permission. In Bitcoin protocols [22], blocks are rejected if the timestamp is later than two hours in the future or earlier than the median time of the last 11 blocks. Therefore, Bitcoin blocks are mined asynchronously but actually accepted in a synchronized manner. To achieve consensus, it includes the first phase to collect the data and the second phase to interpret the data as some proof. In blockchain, both phases need to be decentralized in order to be trustworthy. Other requirements such as efficiency and scalability are also critical for realization. PoW is the first technique used to generate data for collection in reaching consensus.

### Proof-of-Work (PoW)

PoW is a form of cryptographic proof in which one party (the prover) proves to other parties (the verifiers) that a certain amount of a specific computational effort has been expended for reaching some consensus. Verifiers can subsequently confirm this expenditure with minimal effort on their part [23]. PoW is represented by a piece of datum sent from a requester to the PoW service provider. It was first proposed to prevent junk mail [24] by performing some significant computing work before sending an email. Bitcoin was not the first attempt based on PoW [25]. For the first successful hash-based PoW used in blockchain as in Bitcoin, the point of PoW is to prove that enough computing work has been performed so that the winner can append a new block in the blockchain. Some reward could be provided to the winner after the block is really confirmed. Since a block might not be really confirmed due to stochastic finality, the process is called mining and mining nodes are called miners. Mining with reward of bitcoins as in Bitcoin is an incentive for miners to maintain the operation of blockchain, but it is not necessary. The energy consumption problem using PoW does not come from enough work required for PoW but from the competition of obtaining rewards as much as possible. The reward becomes very attractive because of the jumping prices of bitcoins due to the limited supply of bitcoins and centralized holding. The same principle requiring certain amount of "work" applies

to a proof of space, proof of bandwidth and proof of ownership as well. However, proof of stake is different in that it just needs to provide a more deterministic proof of "wealth", e.g., "age" of coin or how long a coin has been created [16].

Each block is normally identified by the hash value of its header. The ID of the previous block is in the header of each block forming a hash chain. The hash chain provides the immutability of blockchain but trust is still not obtained because the incorrect information hashed is still not trustworthy. The previous block confirmed is called the parent block, and the parent of the previous one is called the grandparent block. The value of the cryptographic hash used in blockchain is commonly assumed to be unique and random. Nodes mining with the same parent forms a partition. Bitcoin wins trust because the blocks mined are "randomly" and conducted by hash-based PoW in a decentralized manner. If blocks numbering more than one are mined, broadcasted and then received in different nodes, including self-mined, only the first verified block (still random chosen) is confirmed, with the ancestor blocks and other information corrected if they are not consistent. Then, it is appended as a parent block for mining in each node, possibly creating or merging partitions. The longest chain or main chain is the chain of blocks in each partition built with the most hashes or the largest difficulty accumulated since genesis. Most of the time, only one main chain is the longest. A Bitcoin block is mature if it has 100 blocks confirmed after itself. Only the mature blocks in the main chain can spend their own reward. In this manner of decentralization and partition tolerance, no other consensus mechanism ever provides randomness as well as trust better than Bitcoin. Even if there were Bitcoin related attacks, the Bitcoin core itself has only one split and is restored soon in 2013 due to an undiscovered inconsistency between two versions [26].

### 2.3. Estimable Proof-of-Work (EPoW)

Based on PoW, EPoW records the highest and lowest hash values to estimate how many trials of nonces in each mining as an indicator of estimated computing power of the individual mining nodes. By EPoW, which is US patented [27], the computing power of the mining nodes can be estimated so that we can reject the blocks mined from the nodes that have either too low or high computation power. Therefore, the nodes with big computing power can be discouraged relative to reducing energy consumption, or speculation can be also reduced for the nodes with little computing power. Although this might increase the number of nodes with computing power in predefined range or sybil attack, it can be mitigated economically by decreasing the mining reward or imposing mining fee for each node.

### 2.4. Ourchain

OurChain, meant and read as your chain or our chain, is based on Bitcoin with EPoW mining and a smart contract, OurContract. With the estimated computing power of remote nodes by EPoW, deterministic finality can be reached by PowerTimestamp, which provides global event ordering by combining timestamp and estimated computing power together. The block interval can be 2 s and finality can be reached in two block intervals. The transactions per second (TPS) can be up to 3400 under the 100 Mb/s network bandwidth. Advanced EPoW, called GPoW, is in development for better performance and finality. OurChain is open-sourced and Benevolence licensed, supporting autonomy and sharing. Benevolence license is introduced in Section 4. Each OurChain can be an independent economy or joint economy with other OurChains for special reasons such as performance or culture. By being synchronized with OurChain, other blockchains can also support deterministic finality and cross chain after two block intervals.

#### 2.4.1. Ourchain Architecture

OurChain is a blockchain platform for building blockchains on architecture such as those shown in Figure 1, including the core and services above. This blockchain is similar to a trusted distributed real-time operating system. All terms with prefix "Our" can be

replaced for the new customized blockchain as a variant of OurChain. The core of OurChain is built on top of the Bitcoin core to make use of the existing or coming Bitcoin features such as segwit, lightning, and taproot. OurPoW is the consensus mechanism for better performance and reaches deterministic finality, called OurFinlity. OurSharding and OurDB are the scalable sharding and database technologies that are built. OurContract is the smart contract written in a subset of C or C++ for security, called C−. There are built-in smart contracts as a store to support common tokens or special tokens as votes and certificates. Other OurChain Request for Comments (ORCs) are also to be included.

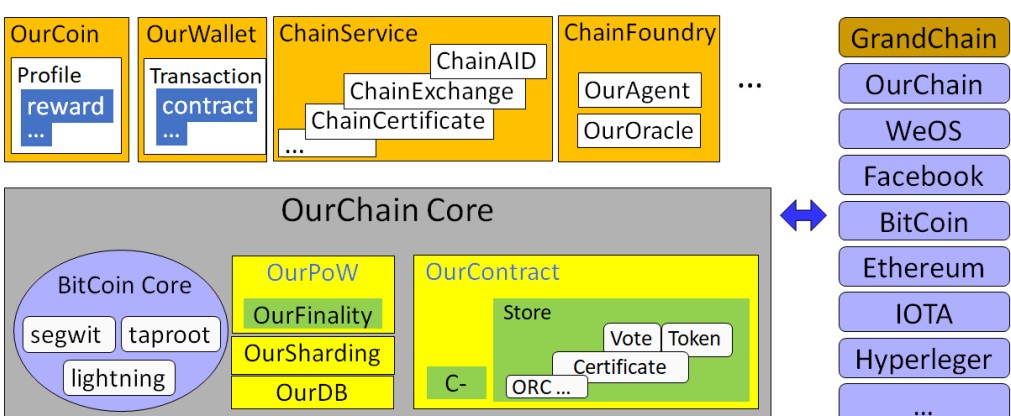

**Figure 1.** OurChain architecture.

On top of OurChain core, there are different services. OurCoin is the built-in payment coin with a profile to customize all OurChain system parameters such as reward for miners. OurWallet is our blockchain wallet for managing cryptocurrencies and sends transactions to exchange information or values by scripts or smart contracts. ChainService provides the service link to other blockchain middleware (DApp), where the information or values are certified and can be cached or shared with other nodes. ChainAID keeps the information of frequent users using AID, an OurChain providing autonomous ID, and can be used to know your customers and conduct Anti-Money Laundering/Combating the Financing of Terrorism (AML/CFT). ChainExchange keeps the latest exchange rates and requests so that it forms a decentralized exchange. ChainCertificate caches the valid certificates recently used. ChainFoundry supports a trusted computing environment for distributed applications (not only DApp), which includes a potential blockchain or oracle testbed. In ChainFoundry, OurAgent coordinates different distributed applications to form a virtual new blockchain, while OurOracle is one of the legal prototyping oracles when some requirement is not fully satisfied, e.g., regulation. An oracle can be a single-node program, a distributed program or a blockchain providing trustful information.

Each OurChain can run on many permissionless or permissioned nodes. Users or OurChain can communicate from core or services with other OurChain or blockchains in the left column of Figure 1. The roles of users or nodes can be certified by AID. GrandChain, a special OurChain, manages the cooperation of OurChains with other blockchains. The cross-chain issues will be discussed in Section 4. WeOS is a degenerated blockchain, where participants are trusted without the unnecessary overhead of private blockchains. Facebook includes the buzzy metaverse to come.

### 2.4.2. Ourcontract

The smart contract in OurChain is called OurContract. It is implemented in C or C++, which is the same as a Bitcoin core, linked by dynamic-link library (DLL) running directly within the OurChain core and not through virtual machines. Therefore, some usage of

"dangerous" functions such as system call needs to be limited. Therefore, we call it "C−". However, it provides the common functionalities of a general operating system such as multi-threading and concurrency. Portability is via the language or the DLL itself instead of bytecode through virtual machines. The virtual machine used in Ethereum [28] is a good method for controlling and measuring how many instructions have been executed to avoid infinite looping and some attacks. However, it is not necessary and timing control can be performed without paying gas. Even running through virtual machines, there are still secure issues on smart contract such as stack overflow. Actually, implementing certification of verified smart contracts is more reliable. Smart contracts need to be customized but are not necessarily rewritten by anyone arbitrarily. The state of each smart contract is stored directly in memory before recording to a file under the directory of the smart contract. To be efficient and scalable, the redesign and integration of these storage into database are necessary when the number of users or sizes of the smart contract codes and the states are relatively big.

Turing-completeness [29] is often discussed on the computer language used to write smart contracts. Actually, most modern computer languages are Turing-complete and so is C−. An imperative language is Turing-complete if it has conditional branching and the ability to change an arbitrary amount of memory. It can perform what Turing machine can perform, but it does not mean that anything can be performed in a smart contract. For example, if the result is time dependent or random, a smart contract cannot be verified at any nodes with the same result. More computing techniques such as PowerTimestamp for distributed synchronization are needed to solve these problems. Moreover, transactions are conducted concurrently in reality. For decentralization, transactions in most blockchains are executed sequentially round-by-round (every block interval) or epoch-by-epoch (certain time period) independently without interaction between transactions to avoid fraud or collusion. Therefore, the synchronization of transactions needs to be performed between confirmations in realization.

*2.5. Powertimestamp*

Since there is no global clock in distributed systems, it is impossible to be sure of the time on a different machine at a different location. That is to say that there is no absolute global event ordering in distributed systems. Only partial ordering or casual ordering is possible, where Lamport has explained in 1978 [8] and proposed a solution called logical clock or Lamport Timestamp [30] later. However, the error of timestamps in a distributed system can be estimated robustly to a given confidence interval [31], even if there are outliers from malicious attackers. With Network Time Protocol (NTP) [32], it can usually maintain time within tens of milliseconds (ms) over the public Internet and can achieve better than one ms accuracy in local area networks under ideal conditions. In addition, asymmetric routes and network congestion can cause errors of 100 ms or more.

Therefore, we can estimate the error above, called TimeError, in a distributed system accurately enough for blockchain down to one-tenth of a second or less, assuming that a dedicated network and flow control were used for the applications that do not tolerate network congestion, such as bank operations. With TimeError, the timestamp of an event and the estimated computing power, as a unique indicator, when the event is issued, a global event ordering can be formed, called PowerTimestamp, such that the event wins if its timestamp is TimeError earlier; otherwise, the event with the higher priority of the indicator wins. That is, if we cannot tell exactly which event is earlier, we decide by a unique indicator. PowerTimestamp is implemented in OurChain such that the TimeError is half of the block interval, 1 s for convenience and the indicator is the estimated computing power by EPoW. The priority of the indicator might be converted so that it is not linear to the estimated computing power for any considerations such as energy consumption, speculation or business model for certain kind of fairness.

With PowerTimestamp, events can be ordered by their issuing time or sending time in the granularity of one-tenth second and the indicator with very fine granularity if users

agree on this difference. Most side effects of network delay can be released. Because Bitcoin protocol needs to check security issues at half of the block interval. A 2 s block interval is set in OurChain to keep all timing manipulation in integers. If the applications need finer timing accuracy, most codes can still work without modification by increasing the hardware clock's rate.

### 2.6. Deterministic Finality

The most important reason why blockchain can provide trust is decentralization, where the artificial randomness from hash-based PoW contributes the most. However, decentralization does not mean no centralization because centralization is necessary sometimes for reasons such as politics, security, privacy or even a centralized clock for synchronization. Actually, decentralization only attempts to be distinguished from centralization as much as possible while avoiding evil (attacks). With PowerTimestamp, distributed synchronization can be achieved. However, for randomness and decentralization, it is not trivial to reach deterministic finality in blockchain because there is still the requirement of the consensus to be met.

In Bitcoin, people believe that 51 attack, one node with more computing power than 50% of the entire network, might change the longest chain so that only stochastic finality can be reached. As mentioned earlier, in Bitcoin protocol, blocks are "synchronized" and rejected if the timestamp is later than two hours in the future or earlier than the median time of the last 11 blocks. That is to say that by following the protocol, only recent blocks can be confirmed and the longest chain still might change, especially when the second longest chain that is catching up is only a few blocks behind. This is because the priority in the consensus mechanism takes the longest chain. In the long run of chains accumulated in steps from now, nothing is impossible, especially if it happens to accumulate nothing in any synchronized steps of any chain branch, i.e., the current longest chain does not grow relatively. OurChain changes the priority of reaching consensus to be the earliest PowerTimestamp, i.e., the earliest timestamp and the highest indicator to avoid the future uncertainty. Even PowerTimestamp can avoid future uncertainty, and the uncertainty from network package loss, system failure or attacks still creates partitions. That is to say that the blocks received in each node might not be the same. We need to merge these partitions while keeping the randomness and fairness of PoW.

The highest or main partition is the partition with the highest priority such as the most nodes or the same most nodes with the largest parent ID. To tolerate partitions and merge them as soon as possible, the new mined blocks with different grandparent are rejected. In addition, the nodes that are not in the highest partition might be lost and need to rejoin the highest one in the next round. Otherwise, it needs to join as a new node. Since the computing power of all nodes can be estimated and constrained, the timestamp of first mined block can also be estimated or refer to the past. Each node starts mining at the next starting time of a block interval since genesis or last block confirmed. Thus, blocks that are issued too earlier might be malicious and can also be rejected. Each node individually samples from the new mined blocks to estimate the highest partition. The mined block in the estimated highest partition with the highest priority by PowerTimestamp, called the highest block, is confirmed and selected as a parent for mining next. It is simple and efficient but might not be accurate. Since the probability to mine a block is proportional to computing power, the estimated highest partition in a node is only likely the highest partition in the entire network. If the estimation is not correct, the node creates a new partition in the next round of mining. It will soon find itself out of the main partition in two rounds because each node needs to have the same grandparent. Therefore, it is suggested (but not necessary) that the system possess more than three new mined blocks to estimate the highest partition. This can be performed by adjusting the target value, which is an indicator of difficulty in mining. If there happens to be no new block mined, a void round with no block can be constructed, forming a void parent but still with a unique ID for next round. The target value needs to be adjusted in a decentralized manner in a short period of

time. Bitcoin is adjusted every 2016 rounds, around two weeks. OurChain can be adjusted much faster or dynamically according to loading.

Any node can probe neighboring nodes to help estimating the highest partition, especially for new nodes or when nodes have high partitions with close sizes. The question of how and when to probe can be adjusted as system parameters for performance or as system features to reach different kinds of deterministic finality. Another attack is to fake timestamps. By adding a hash chain of timestamps for hash trials into PowerTimestamp and imposing time checking on neighboring nodes, timestamp-cheating mined blocks can be detected and rejected. This is another tradeoff and system feature. We assume that most nodes are honest and would like to follow the consensus mechanism to maintain and merge to the main partition wherever possible. Even if there are a lot of malicious nodes executing Sybil attacks, the system state will be still final after two block intervals. The highest block is still random in the sense that the highest indicator is based on a hash-based PoW.

## 3. General Proof-of-Work (GPoW)

Although EPoW provides closed-form formula to estimate the computing power of remote nodes and support deterministic finality, the computation is still complex for further analysis. GPoW extends EPoW generalizing PoW with mathematic model for mining, constructing simple closed-form formula for further trust analysis in blockchains. Using GPoW, at least m valid nonces need to be collected but only m ones, corresponding to the lowest m hash values for estimating computing power, and they are broadcasted for estimating computing power at each round. Users can collect more than m valid nonces to find better m ones to broadcast according to different reward policies. If exactly m valid nonces are collected, it is called a conservative GPoW; otherwise, it is called an aggressive GPoW, as shown in Figure 2. By the definition of order statistic, the lower order statistic has the higher priority of indicating computing power, since the order statistic is not greater than a variable, which can be the target value, and the larger the range from the target value, the greater the computing power it indicates. In other words, with higher computing power to try more nonces, the range would become larger. The other limitation is the time spent for collection. It lowers the priority of PowerTimestamp. If we make the target value large enough so that it is easy to find a valid nonce, much energy can be saved and the randomness of hash-based PoW can still remain. With the first m nonces exactly and not selecting m ones later, the estimated coefficient of variation of estimated computing power can be $CV = \frac{m+1}{m^2+m+1}$, which is explained in the following subsections. The trust indicator of this estimation can be 1-CV. That is, only $m = 10$ trials, and the trust indicator is about 90.09%. If $m = 1000$, it can reach 99.9%. Actually, the general PoW is a proof of PowerTimeStamp (PoPT), depending on the global event ordering constructed by PowerTimestamp using GPoW.

### 3.1. Mining

The reason why Bitcoin uses hash-based PoW for mining to provide trust is that the hashing provides good randomness and fairness (with respect to computing power) to select mined blocks to win mining in a decentralized manner. The trustless trust is more trustful than just immutability or other blockchain features. Each mining node hashes the block header with different nonces until the hash value is not greater than the target value, then, the block is broadcasted for verification and confirmation. Mining, verification and confirmation are all conducted independently from other nodes. For mining a block, the hash values are integers that are uniformly distributed in $[0, T = 2^{256})$. The value $2^{256}$ is included and normalized to standard continuous uniform distribution [0,1] for easier analysis later using a beta distribution. As shown in Figure 2, T is the exclusionary upper bound of hash values, t is the normalized target value, n is the total number of hash trials, m is the number of required valid nonces and $m'$, which is bigger than or equal to m, is the number of total valid nonces. Following the notions in Order Statistics, $X_k$ is the hash value of the kth trial and $X_{(k)}$ is the kth large hash value. The notions i and j were used in

EPoW and $X_{i=(1)}$ is the smallest hash value and $X_{j=(n)}$ is the largest hash value. Therefore, $X_{n=i=(1)}$ is the last trial, with the smallest hash value. The notions *i* and *j* will be replaced hereafter by (1) and (*n*), respectively.

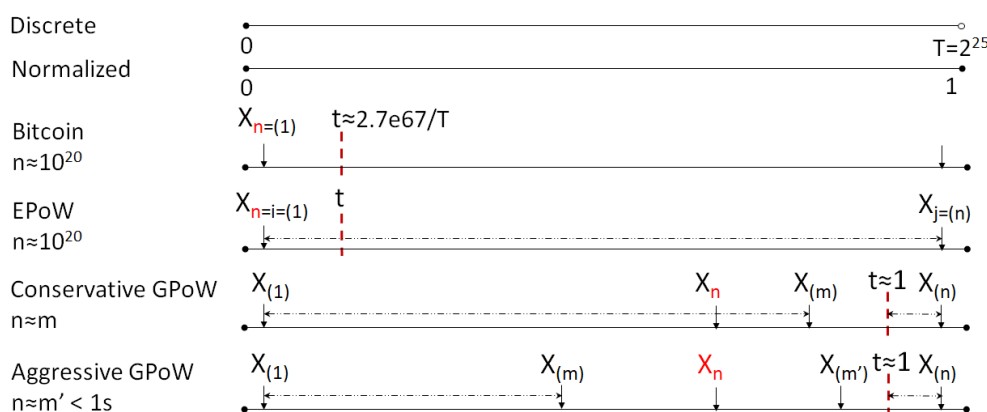

**Figure 2.** General proof-of-work.

### 3.1.1. Mining with the First Nonce

Ignoring the difference of the extra hash value $2^{256}$, GPoW mining with only the first nonce is the same as Bitcoin mining in the first phase of consensus. Since m = 1, the trust indicator is only about 33%. This is why Bitcoin is still not trustworthy enough for formal applications such as CBDC.

### 3.1.2. Mining with the Lowest Two Nonces

As shown in Figure 2, EPoW mining is the same as Bitcoin to find a valid nonce, where the hash value is the lowest, while keep the invalid nonce has the highest hash value. If we redefine the range of the highest hash value, $X_{j=(n)}$ and the lowest hash value, $X_{i=(1)}$, to be a sample range of $X_{(n)} - X_{(1)}$, instead of $X_{(n)} - X_{(1)} + 1$, the new range is still almost the same indicator of computing power. With some variable changes, such as a range of $R = X_{(n)} - X_{(1)}$ and $V = (X_{(n)} + X_{(1)})/2$, the random variable $R$ has a beta distribution, Beta(n−1,2) [17,33], which is the same variance of the second-order statistic Beta(2,n−1). Therefore, EPoW behaves exactly the same as conservative GPoW mining with the second order statistic in the statistical point of view, with the same variance.

### 3.2. Mining with the Lowest M Nonces

Currently, we know Bitcoin and EPoW are only special cases of conservative GPoW. Given any random variables $X_1, X_2, \ldots, X_n$, the order statistics of $X_{(1)}, X_{(2)}, \ldots, X_{(n)}$ are also random variables, defined by sorting the values of $X_1, X_2, \ldots, X_n$ in increasing order. For independent and identically distributed random variables, i.i.d., the order statistics of the uniform distribution on the unit interval have marginal distributions belonging to the beta distribution family [17,18]. It is reasonable to assume that the hash values are i.i.d. For the example of $X_1, X_2, \ldots, X_n$, with a cumulative distribution function (CDF), $F_X(x)$, and probability density function (PDF), $f_X(x) = F'_X(x)$, the order statistics for that sample have a CDF as follows:

$$F_{X_{(m)}}(x) = \sum_{i=m}^{n} \binom{n}{i} (F_X(x))^i (1 - F_X(x))^{n-i}, \tag{1}$$

and the corresponding PDF may be derived from this result by differentiating on *x*, and it is found to be the following.

$$f_{X_{(m)}}(x) = m \binom{n}{m} f_X(x)(F_X(x))^{m-1}(1 - F_X(x))^{n-m}. \tag{2}$$

That is to say that $F_{X_{(m)}}(x)$ is the probability of the event where there are at least m variables in $X_1, X_2, \ldots, X_n$ that are not greater than $x$; more precisely, we have $F_{X_{(m)}}(x) = P(X_{(m)} \leq x)$. This is exactly the same probability of at least $m$ hash values not greater than target $x$ in $n$ trials, which is an aggressive GPoW. Therefore, we can borrow the formula from order statistic for GPoW mining. We denote $U_k = F_X(X_k)$ to obtain the corresponding random sample $U_1, U_2, \ldots, U_n$ from the standard uniform distribution. The order statistics also satisfy $U_{(i)} = F_X(X_{(i)})$. Let $F_U(u) = u$ for the standard uniform distribution; thus, $f_U(u) = 1$. The PDF of the mth order statistic $U_{(m)}$ is equal to the following.

$$f_{U_{(m)}}(u) = m \binom{n}{m} u^{m-1}(1 - u)^{n-m}. \tag{3}$$

That is, the mth order statistic of the uniform distribution is a beta-distributed random variable. Actually, $U_{(m)} \sim Beta(m, n - m + 1)$. Interestingly, it is also the probability of conservative GPoW. By assigning the target value t to u, normalized GPoW mining with at least m valid nonces behaves the same as $U_{(m)}$. Therefore, the statistic formula in beta distribution can be directly applied to GPoW mining. By order statistics, GPoW mining can be mathematically described. For conservative GPoW, once m valid nonces are collected, it stops mining and broadcasts the block for confirmation. However, before it is too late to send out the mined block, miners in nature would aggressively collect more valid nonces for better rewards. For blockchain designers, they need to select an m value and adjust the target value (or difficulty) carefully so that users can trust the blockchain no matter what other users might configure for their system parameters autonomously, or they even manipulate the system maliciously.

### 3.3. Closed-Form Formula of Trust

By the formula above borrowed from order statistic, we derive the following formula. Closed-form formulas for a system are necessary in order to accurately know and design the system. By Formula (3), at the nth trial with target t, the PDF of the mth order statistic $U_{(m)}$ is the folowing:

$$P(U_{(m)}|n) = m \binom{n}{m} u^{m-1}(1 - u)^{n-m} \tag{4}$$

and, by beta distribution, the mean is $E_{U_{(m)}}(u) = \dfrac{m}{n+1}$, and the variance is $V_{U_{(m)}}(u) = \dfrac{m(n - m + 1)}{(n+1)^2(n+2)}$. Since, by using the mth order statistics model, GPoW has "at least" m valid nonces, we can include all trials aggressively, assuming that all trials are independent for simplicity so that the probability of the random variable, the estimated number of trials and N at any trial $i$, $P_N(i)$, can be a constant, $\dfrac{1}{n}$, and is uniformly distributed as used in EPoW. Note that $P_N(i)$ can be complicated depending on user behavior. The marginal probability of $U_{(m)}$ on N is the following.

$$P_N(U_{(m)}) = \sum_{i=m}^{n} P(U_{(m)}|i)P_N(i) = mn^{-1}t^{m-1}\sum_{i=m}^{n} \binom{i}{m}(1 - t)^{i-m} \approx \frac{m}{nt^2}. \tag{5}$$

As in EPoW, by Bayes' theorem, the probability of the ith trial at the mth order statistic of $U$ is the following.

$$P(i|U_{(m)}) = \frac{P(U_{(m)}|i)P_N(i)}{P_N(U_{(m)})} = \binom{i}{m}t^{m+1}(1 - t)^{i-m}. \tag{6}$$

Note that even $P_N(i)$ is set to be simple for simple user behavior, and the simplification for the series of summation with binomial coefficient $\binom{i}{m}$ is not trivial. Refer to Appendix A for more details. Therefore, the mean number of trials is $E_{U_{(m)}}(N) = \dfrac{m+1}{t} - 1$, and variance is $V_{U_{(m)}}(N) = \dfrac{(m+1)(1-t)}{t^2}$. The coefficient of variation (CV) [34] $= \dfrac{standard\,deviation}{mean} = \dfrac{\sqrt{variance}}{mean}$ can be used as an indicator of trust because the standard deviation of data must always be understood in the context of the mean of the data. In contrast, the actual value of CV is independent of the unit in which the measurement has been taken; thus, it is a dimensionless number. For comparisons between data sets with different units or widely different means, one should use the coefficient of variation instead of the standard deviation. In particular, when the data are unique hash values because CV is invariant to the number of replicates, the certainty of the mean improves with an increasing number of replicates. Therefore, the CV of the mth order statistic $CV_{U_{(m)}}(u) = \sqrt{\dfrac{n-m+1}{m(n+2)}}$ and the corresponding CV of the estimated number of trials is $CV_{U_{(m)}}(N) = \dfrac{\sqrt{(m+1)(1-t)}}{m+1-t}$. The CVs depend on $n$, $m$ and $t$. We would like to analyze these variables so that the CVs are minimized. By definition, $E_{U_{(m)}}(u) = \dfrac{m}{n+1} \le t \le 1$, $m \le m' \le \dfrac{m'}{t} \le k \le n$, where $m'$ is the total number of valid nonces and $k$ is an integer. It can be estimated by the deadline, at most $k$ hashes, of sending a block or can be measured in practice. $\dfrac{m'}{t}$ is the total number nonces estimated, including invalid ones. Since we do not know how aggressive the users are and both CVs increase with n and decrease with $t$ and $m$, we can assign $t = \dfrac{m}{m+1}$ and $n = m$ to bound both CVs. Thus, we have the following:

$$CV_{U_{(m)}}(u) = \sqrt{\frac{n-m+1}{m(n+2)}} \ge \sqrt{\frac{1}{m(m+2)}} \tag{7}$$

and the following is the case

$$CV_{U_{(m)}}(N) = \frac{\sqrt{(m+1)(1-t)}}{m+1-t} \ge \frac{m+1}{m^2+m+1}. \tag{8}$$

Both CVs are close in value, around $m^{-1}$, as shown in Figure 3, where $C_u(m)$ and $C_N(m)$ stand for trust indicators $1 - CV_{U_{(m)}}(u)$ and $1 - CV_{U_{(m)}}(N)$, respectively. m can be decided by the smallest integer of m when $\mathrm{CV} \ge \dfrac{m+1}{m^2+m+1}$ to certain percentage such as 0.1%. The initial target value is $t = \dfrac{m}{m+1}$. Then, let the system adjust t periodically so that the mined blocks in a round are sufficient for forming the highest partition easily. More accurate analysis can be performed with more practical data or assumptions. However, since trust indicator 1-CV is between 90% and 99.9% when m is between 10 and 1000, it is enough for most scenarios.

As of 25 December 2021, the Bitcoin network hash rate is $1.77 \times 10^{20}$ H/s [35] running with about 14,764 nodes [36]. Using GPoW aggressively for mining before the deadline, suppose that we use TimeError at one-tenth of a second, at most $k = 10^5$ hashes occur per 2 s block for a block of each PC node with a hash rate of $10^6$ H/s. With the same 14,764 nodes, we can save $\dfrac{1.77 \times 10^{20}}{14,764 \times 10^5/2} \approx 2.4 \times 10^{11}$ folds of energy consumption, let alone conservatively with much less exactly valid nonces per block.

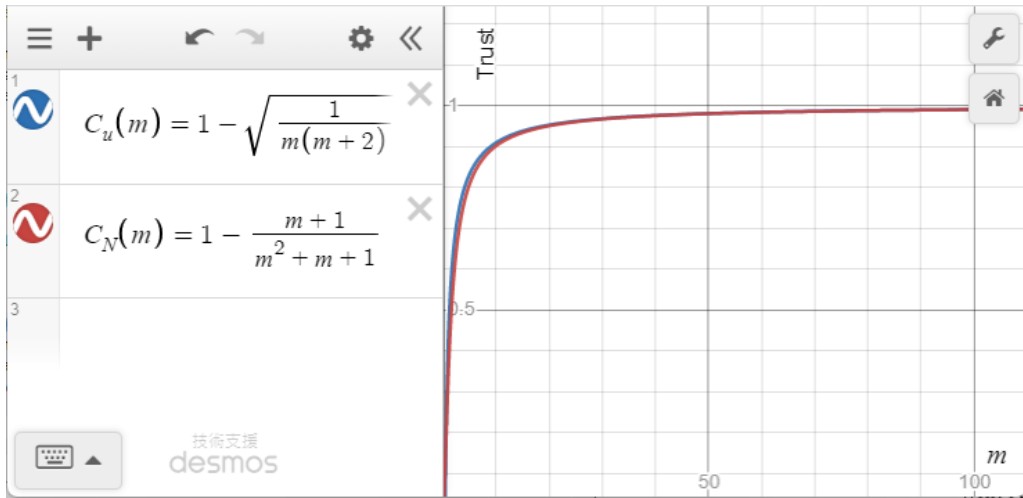

**Figure 3.** Formula of trust.

*3.4. Proof-of-PowerTimestamp (PoPT)*

To reach a consensus, people try many methods for proof, such as work, stake, etc. Nothing can be more fair with justice than time can prove. No chaining is needed, and time is really immutable and there is no way we can change the past. No one can steal time and it cannot be faked. However, we still do not know whether time is centralized or decentralized and how to use time as an instrument to define a global event ordering for us to synchronize distributed events and solve any logic conflicts by ordering. If we can accept the granularity of TimeError in PowerTimestamp with a network timestamp and estimated computing power by GPoW, a global event ordering can be defined by PowerTimestamp and the distributed synchronization for any ordered logic can be possible. Therefore, Proof of PowerTimestamp (PoPT) is one step further than only GPoW, because it involves time checking among neighboring nodes to avoid time faking or network jitter. We can use chaining to avoid fake timestamps as the estimated computing power by hashing, but it might be time consuming. A global protocol with hardware implemented in the network interface card might be a better solution than only using software.

*3.5. Size of Nonces*

Currently, problems come with the number of nonces. In Bitcoin, there is only one nonce with a size of 32 bits used formally in the protocol. Informally, more bits, called extraNonce, can be extended in coinbase, which is the first transaction in a block. With up to a total of 256 bits, which is the same as a hash value, the size for a nonce is enough. However, with much more nonces in GPoW, in addition to the hash chain for secure timestamps, the space needeed for all the nonces is a concern. The nonces can be stored in block header as Bitcoin, with only 80 bytes in the size of the header extended to a couple of kilobytes. Since this information is not very useful after confirmation, we do not need to keep them forever in every block. Keeping the first and the last one in the header for verification after confirmation and the rest separately in another location that is (not every node) less frequently accessed might be a good tradeoff. Since computing power can be estimated and controlled in GPoW, the target value can be tuned so that the computing power can be distinguished by a fewer number of bits in terms of the total nonce size. Actually, in GPoW, the nonce size indicates (is actually) the ratio of the largest computing power over the smallest computing power in the system.

**4. Realization Issues**

With PoPT and distributed synchronization, it appears that most technical problems in blockchain can be solved and can proceed toward realization soon. However, the realization incurs many existing non-technical and new technical problems. Basically, we believe that the trust of one thing is best supported by one blockchain, and many blockchains interacting

on the Internet can discover the disguises of things. For example, the blockchain of traceable agricultural products might not find that some vegetable is as organic as recorded, but a blockchain with fertilizer purchasing might help. Each blockchain is an economy. The economies merge, split or create new ones. We describe some of the problems or issues in the following sections.

### 4.1. Licensing

Open source licensing has been here for many years. It protects intellectual property and encourages innovation. However, currently, it seems to benefit monopolies or authority but benefits less and less for minorities such as individuals or small companies. Because more sources are opened by monopoly or authority, contributions number less for minorities. Blockchain might help to integrate the power of the minorities, but we need a new license for balance. We believe the spirit of blockchain is more on autonomy and sharing than decentralization. With autonomy enforced first, decentralization will follow. Sharing needs to be performed after autonomy; otherwise, it might follow communism. Moreover, if there is not priority, monopoly or authority always win. Therefore, we invent Benevolence Licensing [19]. The regulation is listed as follows:

Benevolence License

1. If you are a natural person, you are free-licensed for personal usage.
2. If you are not a natural person, such as a legal person, group or organization, then you should follow Apache license 2.0, provided that benevolence, neutral dao for humanity and autonomy then sharing, are, by default, supported by a signed contract; otherwise, the name of the violator will be broadcasted.
   (a) Autonomy: Any person involved in this license is free to do anything legally, provided that no one can violate the autonomy of any one.
   (b) Sharing: Any product (not limited to fame, source, object or profit) should be reasonably shared to all participants.
3. If consensus of autonomy or sharing cannot be reached, please quit and leave peacefully by following contracts if they exist. The strife clause is termed for at least one week. Do no evil to any one involved.
   (a) "How to share" needs to be included in this license, including the access to the list of license violators.
   (b) There are no redistribution rights, unless they are explicitly specified in the contract.
4. End of license.

### 4.2. Business Model

If blockchain can support trust on the Internet, before any form of metaverse is realized, we can conduct business on the Internet while governed by blockchains. The business model will be very different and can be implemented in blockchain as well using governance tokens executing online companies or Decentralized Autonomous Organizations (DAOs) [37], including the funding invested with different models for each round. A company might be supported by a blockchain or many blockchains. There might be also many different GrandChain-like NGOs [38]. Wish management experts can provide or conduct some realistic cases soon. The business model is not limited to a commercial one. Art, culture, politics or multidisciplinary ones are the more exciting targets. However, since this is new and multidisciplinary to everyone, trial-and-error is inevitable. More managers or officers in different areas work together in imagining and accepting some digital virtual smart agent or stupid robots built in blockchains that execute human jobs around the clock. Blockchains are numbered 1.0, 2.0, 3.0 or even 4.0 for different stages achieved, similarly to how Web was numbered for decades. Actually, we had defined Web 1.0, 2.0 and 3.0 to be "for-the-people" to read, "by-the-people" to interact and "of-

the-people" to own, respectively. Term Web 3.0 is now borrowed by many people for new applications of blockchain. Before distributed synchronization is realized for complicated business logic, the numbered stages are only marketing prototypes. Each stage might have to be redesigned with distributed synchronization to become "of-the-people", toward the realization as a trusted country-like or UN-like economy. Moreover, sharing the governance tokens with users might be a more effective marketing strategy.

### 4.3. Bank Services

Liquidity-saving mechanism (LSM) [39] can allow payments to be settled with fewer central bank reserves. We need to rethink the role of cryptocurrencies before blockchain can be realized provided that we can perform distributed synchronization in blockchain. Even if there are no technical or performance problems, do we replace traditional currencies with cryptocurrencies completely or to some extent? For example, suppose LSM can be implemented in blockchain, can we issue a check as a security token with an insufficient amount of money in our account? Other bank services such as taxing or interest problems would still need to be rethought as well.

### 4.4. Cross Chain and Database

Cross chain issues and problems become more significant and they still need to be solved in a decentralized manner. However, the related protocols might need to be more similar to HTML instead of CORBA [40], which is learned from the past lessons of centralization. Distributed real-time scheduling methods such as pinwheel [41] can be used to reduce network jitter with more requirements than before because we are transferring values by blockchains. More existing technologies might need to be reviewed in different perspectives. Scalable databases for blockchains need to be redesigned with the similar philosophy of performing cross chain so that resources can be maximally utilized in a global and sustainable point of view.

### 4.5. Universal Distributed Applications

Many distributed applications (DApps) taking advantages of decentralization, the most significant feature of blockchain, would like to explore more market share in existing or even new industries. However, if the applications require no decentralization in nature, most trials fail in a few years. DApps that need to be decentralized are usually universal without limitations of location or situation to run their business. AID and GrandChain are this kind of universal DApps. Frozen chain freezing currencies according to court order or lost chain dealing with the currencies for which its key are lost might also be universal DApps. The question of how to realize universal DApps so that they are sustainable without being a new monopoly is significant in the realization of blockchain.

### 4.6. Sharding

Sharding, parallelizing the execution of blockchain nodes, plays the most significant role in blockchain performance. Since shading or parallelization is data dependent most of the time, the processes of assuming transactions are independent and claiming high TPS is meaningless. Most transactions might be independent from each other. However, if we process transactions in batches or allow LSM, data dependency increases a lot. Therefore, finance and blockchain experts need to cooperate closely to perform transaction flow control in an architectural point of view so that sharding can be successful and provide enough performance for blockchain realization.

### 4.7. Privacy

Privacy is a contradicting feature to the transparency feature of blockchains. Trust could still be provided while preserving privacy in blockchain using the zero-knowledge proof (ZKP) [42] technology. ZKP allows one party to know whether a statement from the other party is true or not with zero private or little public information revealed. However,

ZKP is slow (many minutes per transaction) and the statement is also limited and is not Turing-complete. Since privacy might have higher priority in financing services than performance or transparency, it has to be supported in blockchain with enough computing power before realization. Price-based quantity might be a traditional and effective policy.

### 4.8. Legal Issues

Smart contracts were first proposed in the early 1990s to refer to "a set of promises, specified in digital form, including protocols within which the parties perform on these promises" [43]. Since smart contract can be executed and trusted in blockchain, people would like to execute regulation or even law in blockchain. Therefore, blockchain also stimulates the birth of regulatory (RegTech) [44], legal (LegalTech) [45] and supervisory (SupTech) [46] technologies. These technologies regulate and supervise what can be performed legally in blockchain. They are inevitable subjects toward blockchain realization. As mentioned [13], there exist many myths and misunderstandings among people in different areas. Legislators should be the first ones to understand blockchain without myths.

### 4.9. Attacks

If most things are recorded in blockchains, would any one still like to attack others? Yes, somebody might conduct attacks for fun and do not care about penalties. For example, the Sybil attack might be mitigated by estimable computing power, traceable identity and other techniques so that the attack might only incur little harm in a limited period. However, with the to-be-forgotten right as in GDPR [47], the attacker might perform attacks again by another name easily. Human right, justice, technology and many other related aspects might need to cooperate to solve this problem. However, since the concrete blockchain already provides abstract trust to a certain degree, people can dream of something such as the metaverse. We believe, at some time, blockchain systems can help solve concrete and abstract problems so that any attack is not a big deal and can be controlled. Moreover, we can perform anything with our efforts autonomously, provided that it is not evil with respect to others and there are enough resources that are produced and shared.

## 5. Conclusions

By connecting PoW mining with order statistic, we discover general PoW (GPoW) mining and construct a closed-formula of number of valid trials as a trust indicator of hash-based PoW blockchain. We introduce our previous work, OurChain, based on EPoW mining with deterministic finality and new consensus mechanism Proof-of-PowerTimestamp (PoPT). With GPoW and PoPT, distributed synchronization is feasible and most current blockchain problems can be solved toward realization using OurChain as a blockchain open "foundry". Blockchain is not only a ledger and not a speculative money machine, but it is a trusted operating system for any applications on Internet linked with real world. We only have the prototype of OurChain with EPoW and PoPT, and we extend it with GPoW supporting distributed synchronization toward blockchain realization. Much work remains. We also cannot remove all evil attackers and enforce everyone to be benevolent. However, we believe that the trust indicator of blockchain brings a deterministic direction for everyone to work with deserved reward, and that there is no need for evil. By the Benevolence License, we wish that more individuals will join us in finding a positive position autonomously in order to work, share and live with dignity. There are no more excuses of war from monopoly again!

**Author Contributions:** Conceptualization, C.-W.H. and C.-T.C.; methodology, C.-W.H. and C.-T.C.; software, C.-W.H. and C.-T.C.; validation, C.-W.H. and C.-T.C.; formal analysis, C.-W.H. and C.-T.C.; investigation, C.-W.H. and C.-T.C.; writing—original draft preparation, C.-W.H. and C.-T.C.; writing—review and editing, C.-W.H. and C.-T.C.; visualization, C.-W.H. and C.-T.C. Both authors contribute equally to this research work. All authors have read and agreed to the published version of the manuscript.

**Funding:** This research was funded in part by the Ministry of Science and Technology, Taiwan, grant number 107-2221-E-002-034-MY3 and Ministry of Education, Taiwan, grant number 107L891908, 108L891909, and 109L891809.

**Institutional Review Board Statement:** Not applicable.

**Informed Consent Statement:** Not applicable.

**Data Availability Statement:** Not applicable.

**Acknowledgments:** Thanks for your encouragement: SC Lee, treatment CZ Lee, $\pi$ inspiration CD Chen, supporting TS Hsu and JL Wu and teamwork YL Hsueh.

**Conflicts of Interest:** The authors declare no conflict of interest.

## Abbreviations

The following abbreviations were used in this manuscript:

| | |
|---|---|
| AI | Artificial Intelligence; |
| RPA | Robotic Process Automation; |
| USD | United States Dollar; |
| GDP | Gross Domestic Product; |
| CBDC | Central Bank Digital Currency; |
| PoW | Proof of Work; |
| EPoW | Estimable Proof of Work; |
| GPoW | General Proof of Work; |
| TPS | Transaction Per Second; |
| AID | Autonomous Identity; |
| AML | Anti-Money Laundering; |
| CFT | Combating the Financing of Terrorism; |
| DDL | Dynamic-Link Library; |
| NTP | WithNetwork Time Protocol; |
| CDF | Cumulative Distribution Function; |
| PDF | Probability Density Function; |
| CV | Coefficient of Variation; |
| PoPT | Proof of PowerTimestamp; |
| DAO | Decentralized Autonomous Organization; |
| NGO | Non-Governmental Organization; |
| LSM | Liquidity-Saving Mechanisms; |
| ZKP | Zero Knowledge Proof. |

## Appendix A

In the simplification of formula of series above for means and variances, $\sum_{i=m}^{n} \binom{i}{m} K^{i-m}$, $\sum_{i=m}^{n} i \binom{i}{m} K^{i-m}$ and $\sum_{i=m}^{n} i^2 \binom{i}{m} K^{i-m}$, $0 < K < 1$. It is not trivial and a new technique, called accumulation, is developed by accumulating the recursive items derived from the items separated by difference. For example, we have the following.

- Let $F(n,m) = \sum_{i=m}^{n} f(i,m) = \sum_{i=m}^{n} \binom{i}{m} K^{i-m}, g(i) = \binom{i}{m}, h(i) = K^{i-m}$,
- Let $\nabla_K f(i) = f(i) - K f(i-1), \nabla f(i) = f(i) - f(i-1)$,
- $\therefore \nabla_K h(i) = K^{i-m} - K K^{i-1-m} = 0, \nabla g(i) = \binom{i}{m} - \binom{i-1}{m} = \binom{i-1}{m-1}$,
- $\nabla_K(g(i)h(i)) = g(i)h(i) - K g(i-1)h(i-1) = \nabla g(i)h(i) + (g(i) - \nabla g(i))\nabla_K h(i)$,
- $\nabla_K(\binom{i}{m} K^{i-m}) = K^{i-m} \nabla \binom{i}{m} + (\binom{i}{m} - \nabla \binom{i}{m})\nabla_K K^{i-m} = \binom{i-1}{m-1} K^{i-m}$,
- $\sum_{i=m}^{n} \nabla_K(\binom{i}{m} K^{i-m}) = \sum_{i=m}^{n} (g(i)h(i) - K g(i-1)h(i-1))$
  $= (1-K)F(n,m) + K(g(n)h(n) - g(m-1)h(m-1)) = \sum_{i=m}^{n} \binom{i-1}{m-1} K^{(i-1)-(m-1)}$
  $= F(n, m-1) - f(n, m-1)$,
- $(1-K)F(n,m) = F(n, m-1) - \binom{n}{m-1} K^{n-m+1} - K(\binom{n}{m} K^{n-m} - \binom{m-1}{m} K^{m-1-m})$
  $= F(n, m-1) - \binom{n+1}{m} K^{n+1-m} = F(n, m-1) - f(n+1, m)$,

- $\because$ if $n$ is big, $f(n+1,m) \to 0$, $F(n,1) = \sum_{i=1}^{n} \binom{i}{1} K^{i-1} \approx \dfrac{1}{(1-K)^2}$,

- $\therefore F(n,m) \approx \dfrac{F(n,m-1)}{1-K} \approx \dfrac{F(n,1)}{(1-K)^{m-1}} \approx \dfrac{1}{(1-K)^{m+1}}$.

Therefore, the following is the case.

$$\sum_{i=m}^{n} \binom{i}{m} K^{i-m} \approx \frac{1}{(1-K)^{m+1}},$$

$$\sum_{i=m}^{n} i \binom{i}{m} K^{i-m} \approx \frac{m+K}{(1-K)^{m+2}}, \text{ and}$$

$$\sum_{i=m}^{n} i^2 \binom{i}{m} K^{i-m} \approx \frac{(m+1)(3K+m-1)}{(1-K)^{m-1}} + \frac{1}{(1-K)^{m+1}}.$$

For more realistic user behavior, a beta distribution might be used to model the probability for a next trial, $P_N(i)$, then $\sum_{i=m}^{n} \binom{i}{m}^2 K^{i-m}$, $\sum_{i=m}^{n} i \binom{i}{m}^2 K^{i-m}$ and $\sum_{i=m}^{n} i^2 \binom{i}{m}^2 K^{i-m}$ might be needed. More details will be discussed in another paper.

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
