# Peer review of "Toward Blockchain Realization"

_fintech, doi:10.3390/fintech1010007_

Round 1
Reviewer 1 Report
Pros
1. The paper clearly expresses the current realization problems encountered by the blockchain.
2. The paper presents a good idea to generalize the original version of blockchain PoW.
3. GPoW potentially has the ability to reduce energy consumption.
Cons:
1. There is a wrong word in the paper. For example, global “even” ordering.
2. Some information or thoughts are given too quickly. It would be better to explain more. e.g., lines 382 to 385.
3. This paper seems to have very few academic references.
Reviewer 2 Report
The paper proposes a solution based on general Proof-of-Work, called Proof-of-PowerTimestamp, to reach distributed synchronization and reduce power consumption to less than one billionth of Bitcoin. However, there is no experiment in the manuscript regarding the evaluation of these features.
The blockchain realization issues are widely known in the literature. Rather, it is necessary to highlight the realization issues for your blockchain and not in general. In the background section, articles that have addressed the problems of blockchain realization and how they have been solved by scholars should be cited. I propose some examples:
Shijie Zhang, Jong-Hyouk Lee, Analysis of the main consensus protocols of blockchain, ICT Express, Volume 6, Issue 2, 2020, Pages 93-97, ISSN 2405-9595, https://doi.org/10.1016/j.icte.2019.08.001.
Sedlmeir, J., Buhl, H.U., Fridgen, G. et al. The Energy Consumption of Blockchain Technology: Beyond Myth. Bus Inf Syst Eng 62, 599–608 (2020). https://doi.org/10.1007/s12599-020-00656-x.
Jon Truby, Decarbonizing Bitcoin: Law and policy choices for reducing the energy consumption of Blockchain technologies and digital currencies, Energy Research & Social Science, Volume 44, 2018, Pages 399-410, ISSN 2214-6296, https://doi.org/10.1016/j.erss.2018.06.009.
Where are the Discussions and Conclusions sections?
Reviewer 3 Report
The author explored some of the reasons why blockchain technologies have not yet reached their full potential. The authors presented Proof-of-Power Timestamp, a technique based on generic Proof-of-Work, to achieve distributed synchronization and minimize power usage to less than one-billionth of Bitcoin. This paper required major revision to reach the publication level of this journal. Issues and comments are suggested to the authors.
- For readers to quickly catch the contribution in this work, it would be better to highlight major difficulties and challenges, and your original achievements to overcome them, in a clearer way in the abstract and introduction.
- The literature must be strongly updated with some relevant and recent papers focused on the fields dealt with in the manuscript. Most of the references used in the article are online websites/blogs.
- Authors should provide the comments of the cited papers after introducing each relevant work. What readers require is, by convincing literature review, to understand the clear thinking/consideration why the proposed approach can reach more convincing results. This is the very contribution from the authors. In addition, authors also should provide more sufficient critical literature review to indicate the drawbacks of existing approaches, then, well define the mainstream of research direction, how did those previous studies perform? Employ which methodologies? Which problem still requires to be solved? Why is the proposed approach suitable to be used to solve the critical problem? We need more convinced literature reviews to indicate clearly the state-of-the-art development.
- The authors must discuss the steps of a blockchain transformation journey in order to sequence investment opportunities into a strategic roadmap and short-term action plan.
- The authors must define relationships between the key elements of blockchain metamodel.
- What makes the proposed method suitable for this unique task? What new development to the proposed method have the authors added (compared to the existing approaches)? These points should be clarified.
- Many equations have been presented in the current study. Are these equations formulated by authors or extracted from other studies? Explain. If authors used these equations from other studies, please cite.
- Many Equations are not numbered? Numbering all equations makes it easier for the reader to find a specific equation.
- How does the fintech Chief Technology Officer Assess and understand the potential impact of blockchain on organizations? The absence of criticism and discussion?
- The visibility of the tables and images is not good. Improve the quality of the images.
- No conclusions in the paper?
Round 2
Reviewer 2 Report
The authors addressed all my comments.
Reviewer 3 Report
The authors addressed all my comments.